# Physicochemical Properties of Organic Molecular Ferroelectric Diisopropylammonium Chloride Thin Films

**DOI:** 10.3390/nano13071200

**Published:** 2023-03-28

**Authors:** Ahmad M. Alsaad, Qais M. Al-Bataineh, Issam A. Qattan, Ihsan A. Aljarrah, Areen A. Bani-Salameh, Ahmad A. Ahmad, Borhan A. Albiss, Ahmad Telfah, Renat F. Sabirianov

**Affiliations:** 1Department of Physics, Jordan University of Science & Technology, P.O. Box 3030, Irbid 22110, Jordan; 2Leibniz Institut für Analytische Wissenschaften-ISAS-e.V., Bunsen-Kirchhoff-Straße 11, 44139 Dortmund, Germany; 3Department of Physics, Khalifa University of Science and Technology, Abu Dhabi P.O. Box 127788, United Arab Emirates; 4Nanotechnology Center for Scientific Research, The University of Jordan, Amman 11942, Jordan; 5Department of Physics, University of Nebraska at Omaha, Omaha, NE 68182, USA

**Keywords:** diisopropylammonium chloride (DIPAC) structural, optical, electrical, FTIR, SEM, electric polarization, ab initio calculations

## Abstract

We fabricated ferroelectric films of the organic molecular diisopropylammonium chloride (DIPAC) using the dip-coating technique and characterized their properties using various methods. Fourier-transform infrared, scanning electron microscopy, and X-ray diffraction analysis revealed the structural features of the films. We also performed ab-initio calculations to investigate the electronic and polar properties of the DIPAC crystal, which were found to be consistent with the experimental results. In particular, the optical band gap of the DIPAC crystal was estimated to be around 4.5 eV from the band structure total density-of-states obtained by HSE06 hybrid functional methods, in good agreement with the value derived from the Tauc plot analysis (4.05 ± 0.16 eV). The films displayed an island-like morphology on the surface and showed increasing electrical conductivity with temperature, with a calculated thermal activation energy of 2.24 ± 0.03 eV. Our findings suggest that DIPAC films could be a promising alternative to lead-based perovskites for various applications such as piezoelectric devices, optoelectronics, sensors, data storage, and microelectromechanical systems.

## 1. Introduction

Recently, molecular ferroelectrics have emerged as a promising alternative to traditional inorganic ferroelectrics due to several advantageous properties, such as multi-functionality, low density, low cost, and solution processability. These properties make them potential candidates for the development of all-organic electronic devices. Ferroelectric compounds are characterized by exhibiting ferroelectric–paraelectric phase transitions. Such transitions occur at critical temperatures (*T*_c_) [1]. Conventional inorganic polar crystals such as lead zirconate titanate and barium titanate have been used for decades, owing to their extraordinary ferroelectric properties. However, because of their toxicity effects on the environment and hard processing in addition to their heavyweight, manufactured molecular ferroelectric crystals attract more attention due to their appealing and brilliant properties, such as being lightweight, easy to process, and environmentally friendly [2,3]. However, the practical applications of molecular ferroelectrics have been limited due to their relatively low melting and Curie temperatures, as well as their small spontaneous polarization. To address these limitations, recent studies have investigated the ferroelectric properties of diisopropylammonium halide (DIPAX, X = Cl, Br) molecular crystal systems, which have demonstrated enhanced ferroelectricity [4,5,6].

As a ferroelectric material, DIPAC has a strong tendency for polarization switching, enabling it to be employed for several applications, such as nonlinear capacitors, pyroelectric, data storage, and electro-optical devices [7,8]. DIPAC has been reported to be an inexpensive and easy-to = prepare organic salt with Curie temperature (*T*_c_ = 440 K) [9,10,11,12,13,14,15,16,17]. It exhibits a large spontaneous polarization (*P_s_*) of about 8.9 μC/cm^2^. This value is extremely large compared to those of poly(vinylidene difluoride) (PVDF, ≈8 μC/cm^2^) and Nylon-11 (≈5 μC·cm^2^).

In a previous study, we conducted both computational and experimental investigations of the structural, optical, electronic, crystallographic, and physical properties of thin films composed of diisopropylammonium bromide (C_6_H_16_BrN, DIPAB). Ab initio simulations were also implemented to calculate the key structural parameters. as well as the bandgap energy of DIPAB. The measured and calculated electronic and optical properties of the DIPAB thin films reveal a fairly good agreement between the measured and calculated parameters [1]. In addition, the optical properties of synthesized DIPAB thin films were measured and interpreted. The study was the first of its kind. Previous works were geared toward investigating and interpreting the interplay between the electrical and dielectric properties of DIPAB films. The significance of investigating the optical properties of thin films composed of diisopropylammonium halides (DIPAX, X = Br, Cl) lies in their potential use in a range of applications, including optical lenses, display panels, and solar cells [18].

Few attempts to grow thin-film-based DIPAX (X: F, Cl, Br, and I) have been made on surfaces and typically end up with the formation of randomly distributed microcrystals [19,20]. A way to extend the application of these materials in devices is to control their spontaneous tendency to crystallize in order to obtain homogeneous thin films [21]. The main objective of this study is to optimize the optical and electrical properties of ferroelectric thin films made from diisopropylammonium chloride (DIPAC) that are both cost-effective and high-performing. Successful optimization of these key parameters ensures the scaled fabrication of multifunctional optoelectronic devices that are low cost and that operate more effectively than current devices. In particular, we focus on investigating and interpreting the structural, optical, electronic, crystallographic, and physical properties through experimental means. By doing so, we aim at monitoring the potential characteristics of these materials. 

In parallel, first-principle simulations were conducted to mimic DIPAC thin films to support the experimental investigations [1]. DIPAC has attracted the attention of researchers as a novel organic ferroelectric material. owing to its promising ferroelectric characteristics such as high spontaneous polarization (*P_s_* ≈ 8.9 μC/cm^2^) at room temperature and high-density organic ferroelectric RAM (FeRAM). It exhibits extremely high Curie temperatures (*T*_c_ = 440 K) [5,7]. Therefore, fabricating high-density FeRAM based on ferroelectric DIPAC thin films becomes possible. DIPAC has been extensively investigated for its promising ferroelectric characteristics using various techniques, such as dielectric spectroscopy and X-ray diffraction, to analyze its electrical and structural properties [22,23,24,25]. In this study, we focus on the optical properties of DIPAC thin films as potential candidates for the development of multifunctional scaled optoelectronic devices.

## 2. Experiments and Calculations

The main ingredient used to prepare DIPAC thin films is diisopropylammonium (DIPA) (C_6_H_16_N; Mw = 102.20 g/mol) that was purchased from AK Scientific (Union City, CA, USA). Other materials such as hydrochloric acid (HCl; Mw = 36.46 g/mol) and 12-Crown-4 (C_8_H_16_O_4_) (Mw = 176.21 g/mol) were purchased from Sigma-Aldrich (Darmstadt, Germany). To prepare the DIPAC (C_6_H_16_NCl) solution, 0.999 mL of diisopropylammonium cation and 0.001 mL of hydrochloric acid anion were variegated in 100 mL absolute ethanol in a 1:1 molar ratio by utilizing slow evaporation with continuous magnetic stirring at room temperature. The stabilizer (0.01 mL of 12-crown-4) was then added to the solution while maintaining continuous stirring for 1 h at room temperature. The synthesized DIPAC thin film was coated on a fused silica glass substrate using a dip coating technique for 2 h. The entire mixture was then dried at 40 °C overnight under normal air atmospheric pressure. The thickness of the film was calculated to be about 250 ± 20 nm, using the mathematical model of Al Bataineh et al. [26].

The Fourier transform infrared spectroscopy (FTIR) spectra of the synthesized DIPAC crystalline thin films were obtained using a Bruker Tensor 27 spectrometer (Karlsruhe, Germany) in the spectral range of 4000–400 cm^−1^. Crystalline properties of as-prepared thin films were investigated by measuring X-ray diffraction (XRD) patterns. The patterns were accurately measured by employing a Malvern Panalytical Ltd. (Malvern, UK) diffractometer facility at room temperature with CuKα radiation (0.1540598 nm). The incident angles were varied from 30° to 60° with a step of 0.02° and an energy resolution of 20%. The main objective of the current work was geared toward optimizing the optical and electrical properties of DIPAC thin films.

To elucidate a deeper understanding of the optical properties of the synthesized DIPAC thin films, a UV–Vis spectrophotometer (Hitachi U-3900H, Tokyo, Japan) was utilized to measure the UV–Vis spectra in the 250–700 nm spectral range. Electrical properties were crucial in our investigations of this novel material. The 2D electrical conductivity sheets were obtained at room temperature using a 4-point probe (Microworld Inc., New Jersey, USA) equipped with a high-resolution multimeter (Keithley 2450 Sourcemeter, Beaverton, OR, USA).

To comprehensively deepen the understanding of the structural and physical properties of DIPAC films, ab initio simulations within the framework of the density functional theory [27,28] were utilized to conduct a detailed investigation of DIPAC. The electronic structure was computed by employing the projector augmented wave (PAW) method [29], as implemented in the Vienna ab initio simulation package (VASP) [30]. The Perdew–Burke–Ernzerhof (PBE) exchange-correlation form of the generalized gradient approximation (GGA) [31,32] was implemented to model the layered DIPAC molecule. Moreover, a hybrid functional method [33] based on the Fock exchange in real space was introduced to designate a broad range of molecular properties. The HSE06 hybrid functional method was applied to determine the electronic properties of the polar DIPAC crystal [34,35]. The key parameters, such as optical bandgap, were determined using the HSE06 hybrid functional method. This method was anticipated to yield a more accurate band gap of DIPAC than the values previously reported using the GGA approach (an approach that was proven to underestimate the value of the bandgap). 

It is worth mentioning that the samples prepared in this work can be easily reproducible. The materials used to prepare samples are easily available and can be purchased immediately. The synthesis technique used to prepare samples is straightforward. Indeed, all samples of different sizes and features were reproduced several times. The characterization techniques of the samples employed in this work have shown that different samples yield the same results. Since the samples were subjected to same preparation conditions, the obtained results on different samples were statically the same. This was an adequate indication that the samples were reproducible. Moreover, the materials used, the preparation techniques employed, and the characterization methods utilized were inexpensive and easy to perform.

## 3. Results and Discussion

### 3.1. Chemical Properties

The chemical structure of DIPAC is mainly composed of DIPA molecules bonded together via chloride ions. It exhibits a large value of spontaneous polarization around $P_{s}\sim 8.82\;\mu C\cdot cm^{-2}$ [4]. The FTIR spectra of the DIPAC film (Figure 1) were investigated to study the nature of the interaction between DIPA and chloride ions. The vibrational bands between 2400–3000 cm^−1^ were assigned to N–H stretching vibrations, while the vibrational band at 2095 cm^−1^ was associated with N–H bending vibrations. The -CH_3_ stretching bands appeared at 1585 cm^−1^, while the C–N and C–O stretching bands were located in the 1250–1400 cm^−1^ and 1000–1200 cm^−1^ spectral ranges. The C–Cl stretching band was the main vibrational band that determined the interaction between the DIPA and Cl and appeared in the 500–1000 cm^−1^ spectral range.

### 3.2. Crystal Structure and Morphological Properties

The XRD pattern (Figure 2) of DIPAC film exhibits peaked at 12.24°, 16.78°, 22.46°, 25.04°, 27.42°, 31.98°, and 32.86°, corresponding to DIPAC crystallographic planes indexed by Miller indices ((001), (110), (020), (002), (012), (201), and (300), respectively). The obtained XRD pattern clearly indicated the polycrystalline monoclinic structure of DIPAC molecule.

The monoclinic structure of the DIPAC molecular crystal was also determined by DFT-based calculations, with lattice constants of a=7.495 Å, b=7.818 Å and c=7.655 Å, $\alpha^{{}^{\circ}}=\gamma^{{}^{\circ}}=90^{{}^{\circ}}$ and $\beta^{{}^{\circ}}=114.640^{{}^{\circ}}$ (Table 1, Figure 3). The lattice constants and angle $\beta^{{}^{\circ}}$ were calculated using the following formula:(1)$$\frac{1}{d^{2}}=\frac{1}{\sin^{2}\beta }\left(\frac{h^{2}}{a^{2}}+\frac{k^{2}\sin^{2}\beta }{b^{2}}+\frac{l^{2}}{c^{2}}-\frac{2hl\cos\beta }{ac}\right)$$

The lattice constants (a, b, and c) of the monoclinic DIPAC molecular crystal were determined using the formula presented, where d represents planar spacing and was computed using Bragg’s law with X-ray wavelength λ (0.1540598 nm) and incidence angle θ. The values of the lattice constants of the DIPAC film were calculated and tabulated, as shown in Table 1. The results obtained through DFT calculations were compared to the XRD experimental outcomes, as presented in Figure 2, indicating good agreement.

The crystallite size (D) and lattice strain $\langle \varepsilon \rangle$ were calculated by employing the Williamson–Hall (W–H) method modified by the uniform deformation model (UDM), according to the previous literature [33]. The estimated values of the crystallite size D and the microstrain $\langle \varepsilon \rangle$ of DIPAC film were tabulated, as shown in Table 1.

As shown in Figure 4, SEM micrographs were taken at 50, 10, and 1 µm enlargement scales to observe the surface of DIPAC film. It was found that the film had cracks that were distributed in an island-like pattern. The cracks on the surface provided paths for the penetration of chlorides into DIPA molecules, which led to the reinforcement of chloride-induced island-like patterns. The morphology of the elongated microcrystals was observed to be island-shaped with a common orientation over a large area for all enlargement scales. The width and height distributions of the microcrystals were greatly influenced by growth conditions. The size of the single unit was approximately 100 nm. Additionally, the short-scale micrograph (1 µm) indicated that the coarse units of the cracks also exhibited a micro-sheet-like pattern.

### 3.3. Optical Properties

The optical properties of DIPAC film were examined using a UV-Vis spectrophotometer at room temperature within the spectral range of 250–700 nm. The transmittance spectra showed a rapid increase in values from 0 to 90% as the incident photon wavelength increased from 300 nm to 370 nm, with negligible change in values as the wavelength increased from 370 nm to 700 nm (Figure 5a). The decrease in the transmittance spectra below the absorption edge can be attributed to the interband transition. The band gap energy was determined using Tauc plot [34] and found to be 4.05 ± 0.16 eV (inset of Figure 5a).

The refractive index and extinction coefficient were calculated using transmittance and reflectance spectra, based on previous literature [26,35]. The refractive index exhibited normal dispersion in the (350–700) nm spectral region, with a continuous decrease in values from 1.88 to 1.54 as the wavelength of the incident photon increased (Figure 5b); the refractive index spectra of DIPAC film was affected with the cracks behavior. In addition, the spectral region of (250–350) nm showed anomalous dispersion, due to the resonance phenomenon, occurring when the frequency of the incident photon matched the plasma frequency of the vibrating electric dipoles. The extinction coefficient showed a decrease in the high energy region (300–350) nm and vanished for wavelengths greater than 400 nm, indicating that the DIPAC film was transparent in the visible region (Figure 5c).

### 3.4. Electric Polarization of DIPAC

The Berry phase quantum mechanical approach was used to describe the macroscopic polarization of DIPAC [28,29,36,37,38], and the difference in total polarization between two phases was the spontaneous polarization $P_{s}$. The equilibrium lattice parameters were slightly reduced by performing the Van der Waals correction to energy using the DFT-D3 method with Becke–Jonson damping in VASP [39,40]. According to the calculations, the spontaneous polarization of α-DIPAC was $P_{s}=8.82\;\mu C/{cm}^{2}$, which was consistent with previous theoretical results [41]. 

DIPAC has the highest *T*_c_ among known molecular ferroelectrics (*T*_c_ = 440 K). These values of $P_{s}$ and *T*_c_ suggest that DIPAC has the potential to be used in high-temperature piezoelectric and optoelectronic devices. The energy difference between the polar and paraelectric phase is 12.5 eV, which can be used to determine the Curie temperature of DIPAC crystal [42,43]. The ferroelectric–paraelectric phase transition occurs at 440 K, which is well above room temperature, making DIPAC an excellent alternative to perovskites for high-temperature device applications [44].

The electrical conductivity of the DIPAC film was measured using a 4-point probe at various temperatures ranging from 300 K to 323 K. A conductivity mapping of the DIPAC film (Figure 6a) indicated that the film had a low conductivity with a non-significant distribution, averaging 3.6 μS/cm. The average electrical conductivities (σ) for temperatures 300 K, 308 K, 313 K, 318 K, and 323 K were plotted against (1000/T(K)), as shown in Figure 6b, and fitted to the Arrhenius formula. The Arrhenius-like behavior of σ is described as $\sigma =\sigma_{0}\exp\left(-E_{a}/K_{B}T\right)$, where $\sigma_{0}$ is the pre-exponential factor, T is the temperature [K], $K_{B}$ stands for the Boltzmann constant, and $E_{a}$ is the activation energy [45]. The $E_{a}$ parameter was deduced from the electrical conductivity was found to be 2.24 ± 0.03 eV. The perfect fit indicated that the system is thermally activated. This behavior can be directly related to the significant surge of the cations’ $E_{a}$ value, leading to the cation jump to the next coordinating site and, thus, increasing the energy of segment vibrations [46] which, in turn, leads to abrupt segmental motion counter to the hydrostatic pressure [47]. In addition, increasing the temperature enhances a significant carrier concentration mobility of the free electrons [46,48].

### 3.5. Electronic Properties

We optimized the lattice parameters of DIPAC using a total energy minimization approach and followed the Hellmann–Feynman (HF) forces on the ionic sites in the unit cell of DIPAC. The HF forces were minimized to be as small as 0.002 eV/Å. We calculated other key parameters, such as the density of states (DOS) and the band structure of DIPAC. The electronic band structure of polar DIPAC was computed using the HSE06 method, as demonstrated in (Figure 7a). The calculated bandgap energy was found to be approximately 4.5 eV, indicating that polar DIPAC is a wide-bandgap insulator. The DOS and partial DOS (PDOS) of DIPAC were also calculated using the HSE06 method (Figure 7b). The PDOS plots indicated that the valence band consists mainly of Cl (p) orbitals, while the conduction band is dominant by C(p), C(s), N(p), and N(s) states. This confirms the fact that the interband transitions mainly occurred from the valence bands of Cl atoms to the conduction bands of C and N atoms in the DIPAC molecule.

## 4. Conclusions

In conclusion, as-synthesized DIPAC thin films are primarily composed of DIPA molecules that are interconnected through chloride ions, as revealed by XRD as well as by FTIR measurements. The measured optical bandgap value of DIPAC films indicated that it behaves as an insulator and exhibits excellent dielectric properties. We obtained high-transmittance value in the 300–700 nm spectral range in the DIPAC film. DIPAC exhibited a wide optical bandgap of 4.5 eV. Increasing the temperature enhanced the available free volume around the DIPAC chains significantly. Consequently, higher ion mobility and thermally activated electrical conductivity of DIPAC films were observed in this work. Using the Berry Phase Approach (BPA), polar DIPAC was found to exhibit a ferroelectric phase transition temperature of 440 K with a large value of spontaneous polarization of 8.82 µC/cm^2^. This spontaneous polarization is comparable to those of certain environmentally harmful perovskites, indicating that DIPAC has the potential to be a suitable alternative for high-temperature piezoelectric-based applications. The electrical conductivity of DIPAC was measured and the activation energy was determined to be 2.24 ± 0.03 eV. The system was found to be thermally activated, and this behavior can be attributed to the increase in cation thermal activation energy. This increase triggered the cation to jump to the next coordinating site. This jump induced a significant increase in the segmental vibrations energy, which in turn led to more segment motion to counter the increase in the hydrostatic pressure. Thermal agitations induced by the increase of the temperature enhanced the mobility of the free electrons significantly. Overall, the structural, optical, electronic, and electrical properties of DIPAC thin films measured and interpreted in this work indicate that these films could be potential candidates for the fabrication of a promising new generation of efficient multifunctional optoelectronic devices for a wide range of technological applications.

## Figures and Tables

**Figure 1 nanomaterials-13-01200-f001:**
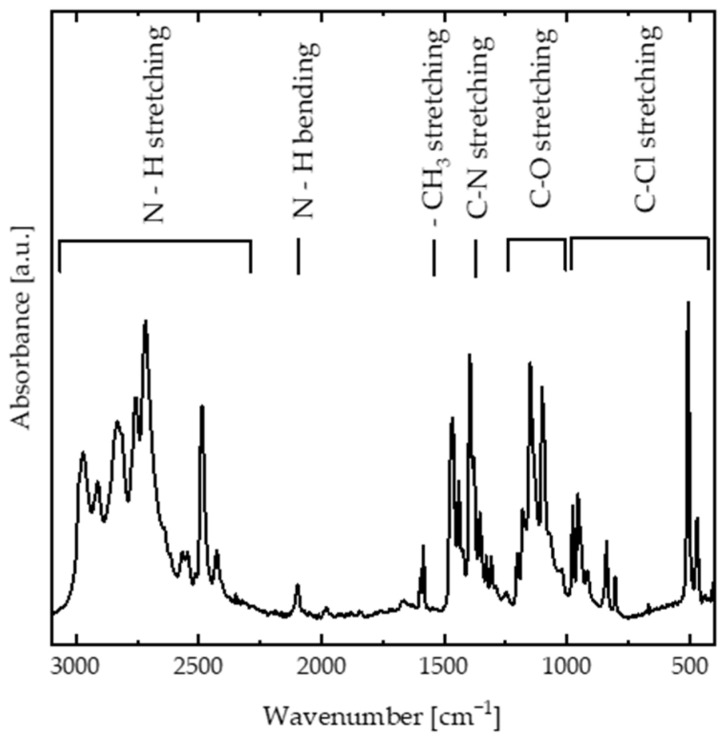
The FTIR spectrum of DIPAC.

**Figure 2 nanomaterials-13-01200-f002:**
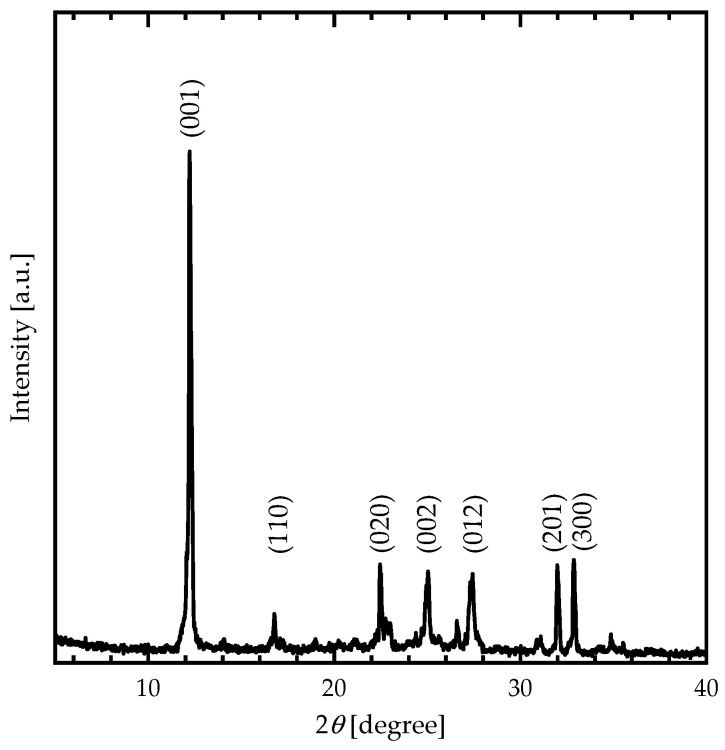
The XRD patterns of the DIPAC thin films.

**Figure 3 nanomaterials-13-01200-f003:**
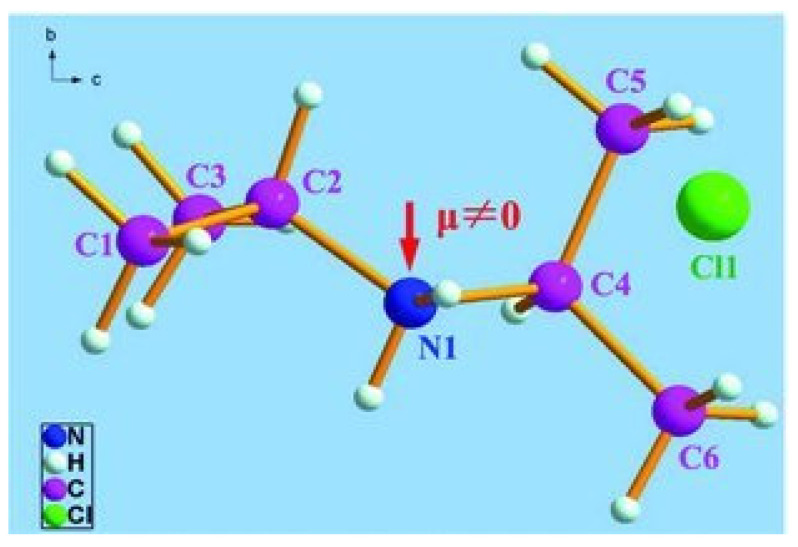
A schematic diagram of the DIPAC polar crystal.

**Figure 4 nanomaterials-13-01200-f004:**
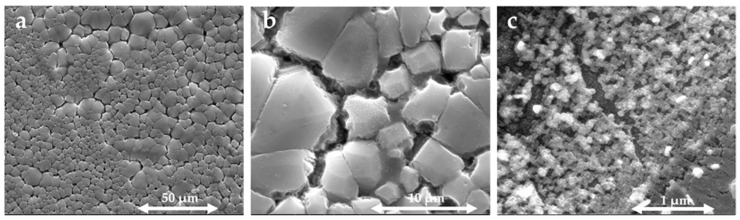
The SEM micrographs of DIPAC film at different enlargement scales (**a**) 50, (**b**) 10, and (**c**) 1 μm.

**Figure 5 nanomaterials-13-01200-f005:**
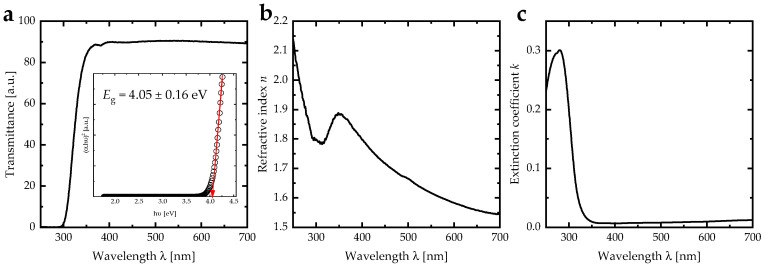
(**a**) Transmittance, (**b**) refractive index, and (**c**) extinction coefficient of DIPAC film. The inset in (**a**) represents the Tauc plot of DIPAC film.

**Figure 6 nanomaterials-13-01200-f006:**
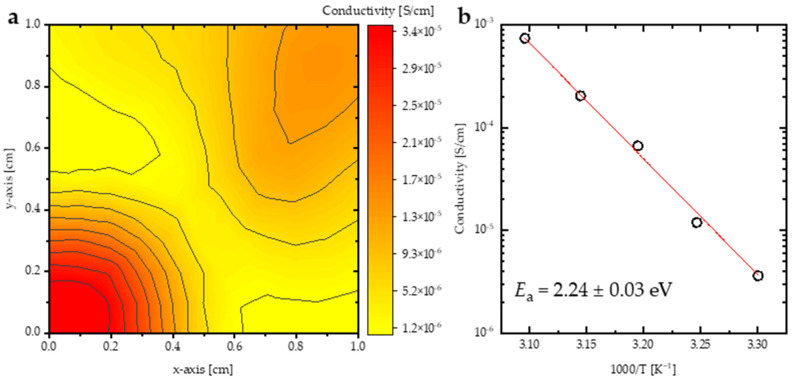
(**a**) The conductivity mapping (1 cm × 1 cm) of DIPAC thin film, (**b**) electrical conductivity of DIPAC films as a function of 1000/T [K^−1^].

**Figure 7 nanomaterials-13-01200-f007:**
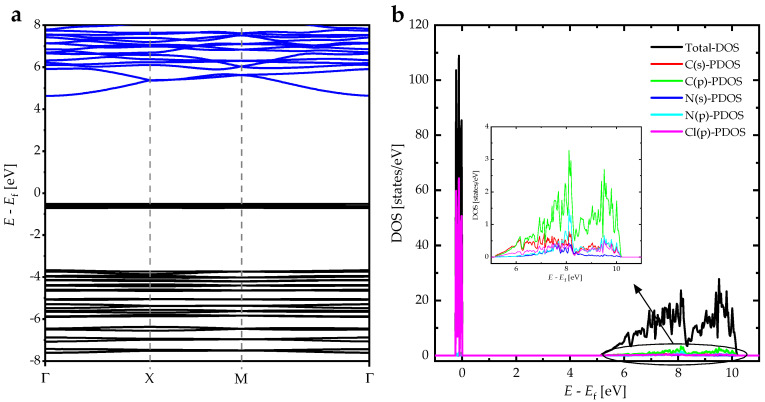
(**a**) The electronic energy band structure, and (**b**) the DOSs and PDOSs of DIPAC.

**Table 1 nanomaterials-13-01200-t001:** The structural properties of polar monoclinic phase of DIPAC molecular crystal.

Parameter	DFT	Exp.
Empirical formula	C_6_ H_16_ Cl N	C_6_ H_16_ Cl N
Polarization [μC·cm^−2^]	8.90	--
Crystal system	Monoclinic	Monoclinic
Space group	P2_1_	P2_1_
Lattice parameter *a* (Å)	7.495	7.239
Lattice parameter *b* (Å)	7.818	7.901
Lattice parameter *c* (Å)	7.655	7.397
$\alpha^{{}^{\circ}}$	90	90
$\beta^{{}^{\circ}}$	114.640	114.870
$\gamma^{{}^{\circ}}$	90	90
Crystallite size (nm)	--	10
Strain	--	0.0095

## Data Availability

The data that support the findings of this study are available from the corresponding author upon reasonable request.

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
