# Peer review of "Physicochemical Properties of Organic Molecular Ferroelectric Diisopropylammonium Chloride Thin Films"

_nanomaterials, 2023, doi:10.3390/nano13071200_

Round 1

Reviewer 1 Report

In this manuscript, the authors synthesized organic ferroelectric films and investigate their optoelectronic properties. Furthermore, the authors attempt to verify their experimental results utilizing DFT calculations. The material shows potential as an alternative to perovskite oxides for low-cost optoelectronic devices, and falls within the scope of Nanomaterials. However, I have found some issues and would recommend publication after major revisions. I also suggest the authors to seek English editing service for clarity of language.

1.       Line 176: “…they play a role in the initiation and propagation stages…”
I do not see any evidence or hypothesis for this claim.

2.       Line 191: “…the absence of free electrons…”
- 370 – 700 nm is not the range where you would see contribution from free electrons.
- the authors also claim that there are free electrons in Line 235, which is inconsistent
Line 192: “…ejection and transportation…”
- I think this is fundamental absorption (i.e. interband transition), and therefore this is not related to free electrons as claimed in the preceding sentence.

3.       Figure 5: units on x-axis should be “nm” not “cm-1

4.       Line 235: “…upsurge of the mobility…”
- isn’t it the carrier concentration that increases with temperature here?

5.       Line 260: “…transmittance value…”
- wrong range and units.

Author Response

The point-point-rebuttal-Reviewer-1 is attached

Reviewer 2 Report

The manuscript nanomaterials-2288751 describes a study about diisopropyl ammonium chloride (DIPAC) deposited on a surface by deep coating.

The topic is exciting and fits the journal's aim. I have only minor remarks listed below. Once the authors have addressed the points listed below, I expect the manuscript to be suitable for publication in Nanomaterials.

Remarks:

The introduction does not mention important literature on DIPA* ferroelectric compounds processability in film and patterned structures DIPA* (see, for example, Langmuir 2017 33, 12859-). This literature, or similar, should be considered in the introduction.

The manuscript is focused on the characterisation of the film properties. However, the film itself is poorly characterised. The authors should add experimental detail about the film morphology (for example, thickness, roughness, etc.).

Does the quality of the film influence the DIPAC properties? I suggest adding a comment.

As admitted by the authors, the quality of the film is limited (it contains a high number of cracks). Can the quality of the film be improved (for example, by thermal or solvent annealing)?

What about reproducibility in different samples? Is the size of objects generated by the cracks (Fig. 4b) reproducible?

I suggest adding a comment about reproducibility and the possible differences between different samples (if present).

Author Response

The point-point-Rebuttal-Reviewer-2 is attached.
